# Single-Stage Peninsular-Shaped Lateral Tongue Flap for Personalized Reconstruction of Various Small- to Moderate-Sized Intraoral Defects: A Retrospective Case Series with Tongue Function Evaluation Using the Functional Intraoral Glasgow Scale

**DOI:** 10.3390/jpm13121637

**Published:** 2023-11-24

**Authors:** Wonseok Cho, Eun-A Jang, Kyu-Nam Kim

**Affiliations:** Department of Plastic and Reconstructive Surgery, Kangbuk Samsung Hospital, Sungkyunkwan University School of Medicine, 29, Saemunan-ro, Jongno-gu, Seoul 03181, Republic of Korea; wonseok26.cho@samsung.com (W.C.); cassisuna.jang@samsung.com (E.-A.J.)

**Keywords:** intraoral defects, free flap, local flap, reconstruction, single-stage peninsular-shaped lateral tongue flap, functional intraoral Glasgow scale, tongue function

## Abstract

Herein, we present our experience using a single-stage peninsular-shaped lateral tongue flap (pLTF) to cover various intraoral defects and confirm the versatile utility and effective application of pLTF in intraoral defect reconstruction. This study included eight cases (six males and two females; average age 60.3 ± 16.9 years) of intraoral defect reconstruction performed by a single surgeon between August 2020 and May 2023 using the single-stage pLTF technique. Electronic medical records and photographs of the patients were collected and analyzed. The functional intraoral Glasgow scale (FIGS) was used to evaluate preoperative and postoperative tongue function. Defect sizes ranged from 3 cm × 3 cm to 4 cm × 6 cm. Notably, all defects were successfully covered with pLTFs, and the flap sizes ranged between 3 cm × 4.5 cm and 4.5 cm × 7.5 cm. The flaps completely survived without any postoperative complications. At follow-up (average, 9.87 ± 2.74 months), no patient had tumor recurrence or significant tongue functional deficits. The mean preoperative and postoperative FIGS were 14.75 ± 0.46 and 14.00 ± 0.92, respectively (*p* = 0.059). Thus, the single-stage pLTF technique is a good reconstructive modality for various small to moderate intraoral defect coverage in selected cases for personalized intraoral reconstruction.

## 1. Introduction

Intraoral defects resulting from oncological tumor resection are a frequent challenge encountered in plastic surgery [1,2,3,4,5]. The selection of an appropriate reconstructive method for each intraoral defect is based on several critical factors, including its size, characteristics, the structures exposed, and the clinical condition of the patient [1,2]. Reconstructive surgeons are responsible for meticulously evaluating these variables to determine the optimal approach for restoring the affected area’s form and function [2,3]. In contemporary microsurgery advancements, free flap reconstruction is the principal modality for covering intraoral defects [2,3,6,7]. Free flaps in intraoral reconstruction yield superior functional and aesthetic outcomes by providing a robust supply of healthy tissue to address the defect comprehensively [2,3,6,7,8,9]. However, the successful implementation of free flap surgery requires several essential components, such as a proficient microsurgeon, specialized microsurgical instruments, and an adequately staffed postoperative care system [2,10].

Furthermore, it is imperative to recognize that free flap surgery may be contraindicated in cases where patients are unable to endure prolonged operative times due to underlying comorbidities, such as cardiopulmonary conditions, cardiac issues, or impaired renal function [2,3,10]. In such instances, when feasible, local flap techniques may present a suitable alternative, particularly for small- to moderate-sized intraoral defects [1,2,3,4,5,11,12,13,14,15,16,17]. In a previous study, we reported a successful case involving the use of a single-stage peninsular-shaped lateral tongue flap (pLTF) for reconstructing an intraoral defect following tumor resection in the buccal area, which encompassed the retromolar trigone (RMT) region [2]. The current investigation presents our broader experience with the single-stage pLTF technique in addressing various intraoral defects. To our knowledge, there have been no similar studies, and this may be the first case series of single-stage pLTF reconstruction for intraoral defects in Korea. This study aimed to underscore the versatile applicability and efficacy of the pLTF method in intraoral defect reconstruction.

## 2. Materials and Methods

### 2.1. Ethical Compliance

This study adhered to the ethical principles in the 1975 Declaration of Helsinki. The Institutional Review Board of Kangbuk Samsung Hospital approved this study (approval number: 2023-08-052). Written consent was obtained from all patients for using their information and images in publicly available open-access publications before conducting treatment procedures and surgeries.

### 2.2. Patient Selection

We enrolled patients who underwent single-stage pLTF reconstruction to address intraoral defects between August 2020 and May 2023. Patients who received intraoral defect coverage using other local or free-flap techniques were excluded. Retrospective data analysis was performed using patients’ electronic medical records and clinical digital photographs. Microsoft Excel (Microsoft, Redmond, WA, USA) was employed for data processing and analysis, ensuring patient anonymity. The data encompassed factors such as the cause of the defect, its location, size, flap dimensions, flap elevation time, flap survival rates, complications, pathologic tumor–node–metastasis (TNM) staging, postoperative adjuvant treatment, preoperative and postoperative tongue function, and follow-up periods.

### 2.3. Surgical Techniques

We utilized a posteriorly based pLTF in all cases. A 3-0 black silk tagging suture was placed at the apex of the tongue to mark the midline groove and facilitate tongue manipulation [2]. On the ipsilateral side of the defect, the pLTF was designed with a length exceeding the mobile tongue length (consistently >5–7 cm in all cases) and a width slightly greater than the defect’s lateral dimension [2]. The flap width was constrained only by the potential for direct donor site closure. The flap, employing an electrocautery device, was meticulously elevated in a full-thickness manner, encompassing a cuff of the intrinsic muscle attached to the mucosal layer to safeguard the mucosa from shear and heat. The flap elevation did not extend beyond the circumvallate papilla to prevent injury to the vascular pedicle [2]. A minimal mucosal bridge was retained at the pivot area of the flap to protect the pedicle, provide structural support during flap transfer, and enhance vascular perfusion [2,18]. If an intact mucosal layer remained between the flap and the defect, it was incised and opened to facilitate flap transfer without hindrance [2].

The direction of flap movement was determined based on the defect’s location. The flap was rotated laterally and advanced upward to transfer to the defect site for defects on the buccal mucosa and retromolar trigone. In cases of mouth floor defects, the flap was rotated medially and advanced downward. For defects around the root of the tongue, the flap was rotated laterally and advanced downward. The final appearance exhibited a peninsula-like configuration following the flap inset into the defect. The donor site was closed directly without tension. Figure 1 shows a schematic representation illustrating the single-stage pLTF reconstruction for intraoral defect coverage in various anatomical areas.

### 2.4. Evaluation of Tongue Function with the Functional Intraoral Glasgow Scale (FIGS)

Pre- and postoperative assessments of tongue function were conducted using the functional intraoral Glasgow scale (FIGS). The FIGS, developed by the Canniesburn Hospital Plastic Surgery Unit staff, is a straightforward self-assessment scale designed to ascertain patients’ ability to chew, swallow, and speak before and after surgery. It comprises three items: the ability to chew, swallow, and speak [19,20]. Each item is rated on a five-point scale, with five indicating no disability and one indicating an inability to chew, swallow, or speak [19,20]. Therefore, the total score can range from a minimum of 3 to a maximum of 15. Patients in this study self-assessed their FIGS status before and after pLTF reconstruction for intraoral defect coverage. The preoperative FIGS was evaluated in the hospital ward one day before pLTF reconstruction surgery. The postoperative FIGS was evaluated in the outpatient clinic at the final follow-up appointment. Individual scores on the five-point FIGS represent the following: for the ability to chew, 5 = “any food, no difficulty”, 4 = “solid food with difficulty”, 3 = “semisolid food, no difficulty”, 2 = “semisolid food with difficulty”, and 1 = “cannot chew at all”; for the ability to swallow, 5 = “any food, no difficulty”, 4 = “solid food with difficulty”, 3 = “semisolid food only”, 2 = “liquids only”, and 1 = “cannot swallow at all”; and for the ability to speak, 5 = “clearly understood always”, 4 = “requires repetition sometimes”, 3 = “requires repetition many times”, 2 = “understood by relatives only”, and 1 = “unintelligible”. We used R language version 3.3.0+ (R Foundation for Statistical Computing, Vienna, Austria) for all statistical analyses. Continuous variables were expressed as mean ± standard deviation (SD). We used Student’s *t*-test for continuous variables to compare the differences between preoperative and postoperative FIGS. The statistical significance level was set at *p* < 0.05.

## 3. Results

Table 1 provides an overview of the clinical data of the patients involved in this study. The study included eight patients (six males and two females), with an average age of 60.3 years (±16.9 years; range, 34–86 years). The causes of defects were as follows: wide local excision of squamous cell carcinoma in five patients, adenocarcinoma in two patients, and adenocystic carcinoma in one patient. The locations of the defects were distributed as follows: buccal mucosa and RMT in three patients, root of the tongue in three patients, and mouth floor in two patients. Defect sizes ranged from 3 cm × 3 cm to 4 cm × 6 cm. Notably, all defects were effectively addressed using pLTFs, with flap sizes ranging from 3 cm × 4.5 cm to 4.5 cm × 7.5 cm. The mean flap elevation time was 28.75 min (±5.17 min; range 20–35 min). Donor sites were primarily closed in all cases. Remarkably, all flaps exhibited complete survival, and no postoperative complications were recorded. Postoperative adjuvant treatments were administered to five patients. At an average follow-up of 9.87 months (±2.74 months; range, 7–15 months), no tumor recurrence or significant deficits in tongue function were observed.

Regarding tongue function, the mean pre- and postoperative FIGS scores were 14.75 ± 0.46 and 14.00 ± 0.92, respectively (*p* = 0.059). Table 2 provides a summary of the FIGS data. In the subsequent section, we present representative cases to illustrate the efficacy of single-stage pLTF reconstruction for intraoral defect coverage.

### 3.1. Case Presentations

#### 3.1.1. Case 1: Intraoral Defect Involving the Root of the Tongue

A 34-year-old male with no prior medical history was diagnosed with squamous cell carcinoma located on his tongue’s left root following a punch biopsy. The preoperative staging indicated no distant metastasis; his initial FIGS score was 15. Our head and neck surgeon performed a wide excision of the lesion, maintaining a safety margin of 5–7 mm, and also conducted an ipsilateral selective neck dissection across levels I, II, and III. Notably, all resection margins were confirmed tumor-free by our senior pathologist using frozen sections. The final defect size was 3 cm × 4 cm, and a posteriorly based pLTF, measuring 3.5 cm × 5 cm, was designed corresponding to the defect’s ipsilateral side and was successfully transferred using a laterally downward rotation and advancement movement (Figure 2A–C). The flap exhibited complete survival post-procedure, and the patient faced no complications (Figure 2D). Pathology reports indicated a TNM stage of pT1N0M0. No additional postoperative treatments were administered. At the 7-month follow-up, no signs of tumor recurrence were detected, and the patient’s FIGS score remained at 15.

#### 3.1.2. Case 2: Intraoral Defect Involving the Mouth Floor

A 62-year-old male, diagnosed with adenocarcinoma on the right mouth floor following a punch biopsy, presented with comorbidities, namely hypertension and hypothyroidism. Preoperative assessments showed no distant metastasis; his FIGS score was 15. Our surgical team performed a comprehensive excision of the lesion, ensuring a safety margin of 10–15 mm, along with an ipsilateral sublingual gland excision and sialodochoplasty. Frozen sections verified all resection margins to be tumor-free. The final defect size was 3 cm × 4 cm, and a 3 cm × 6 cm posteriorly-based pLTF was crafted and aptly transferred to the defect using a medially downward rotation and advancement movement (Figure 3A,B). The flap showed complete viability without any postoperative complications (Figure 3C). Pathologic assessments revealed a TNM stage of pT1N0M0. The patient underwent radiation therapy a month after the procedure, receiving 60 Gy in 30 fractions to mitigate the chances of local tumor recurrence. Seven months post-operation, no tumor recurrence was observed, and his FIGS score was 14 (Figure 3D–F).

#### 3.1.3. Case 4: Intraoral Defect Involving the Buccal Mucosa and RMT

An 86-year-old male with a medical history of hypertension and diabetes mellitus was diagnosed with squamous cell carcinoma in the left buccal mucosa and retromolar trigone area after a punch biopsy. Preliminary staging highlighted no distant metastasis, and the patient’s FIGS score was 14. A wide tumor excision, with an 8–10 mm safety margin, was performed alongside an ipsilateral selective neck dissection spanning levels I through IV. Notably, all margins were confirmed as free of tumor cells. After confirming that the final defect size was 3.5 cm × 4.5 cm, a posteriorly based pLTF measuring 4 cm × 6 cm was designed and transferred to the defect using a laterally upward rotation and advancement method (Figure 4A–D). The flap demonstrated complete survival, with the patient experiencing no complications. The pathologic stage was determined as pT3N0M0. The patient opted against any postoperative radiation therapy. After 15 months, the patient showed no signs of tumor recurrence and maintained a FIGS score of 14 (Figure 4E,F).

## 4. Discussion

We have presented the results of a single surgeon’s experience with intraoral defect coverage using the single-stage pLTF technique in eight consecutive cases. One of the key strengths of the pLTF technique is its technical simplicity and ease of execution. We argue that its relative simplicity streamlines the surgical process, making it accessible to a broader range of surgical teams. This is an essential aspect, as it could increase the availability of effective reconstruction options for patients with intraoral defects. Our favorable outcomes can be attributed to the effective application of the pLTF technique, particularly in preserving good tongue function.

Intraoral defects following oncological surgery pose inherent complexities, making intraoral defect coverage a constant challenge [1,2,3]. As with reconstructions in other anatomical regions, a fundamental principle of intraoral defect reconstruction is the replacement of tissue with a similar counterpart, using the simplest technique available when feasible [2,3]. In this context, local flaps for intraoral reconstruction excel in providing a texture, color, and contour match similar to the surrounding tissue [2]. Notably, intraoral flaps situated within or adjacent to the surgical field offer a mucosal layer coverage comparable to intraoral defects. These flaps, including buccal, palatal, and tongue flaps, are valuable in addressing intraoral defects that fall within the spectrum of size; they are neither small enough to be closed primarily nor large enough to necessitate the use of free flaps [1,21].

Among the array of intraoral flaps, tongue flaps have clear advantages, such as dependable vascularity, versatility due to their central location, tissue redundancy and elasticity, and the simplicity of flap harvesting [2,3,4,5,22]. Tongue flaps have been applied for intraoral defect coverage in various types, including dorsal, ventral, lateral, sliding, and island tongue flaps [3,4,5,21]. Notably, various types of tongue flaps, excluding sliding and island tongue flaps, are primarily employed as interpolation flaps [3,4]. This entails a subsequent operation for pedicle division, typically performed around 3–5 weeks after the initial flap surgery, demanding the patient’s cooperation in maintaining the pedicle and undergoing staged operative procedures [3,4,5]. However, within the spectrum of tongue flaps, posteriorly based dorsal and lateral tongue flaps can be applied with a single-stage flap operation for intraoral defect coverage, as confirmed by several studies [2,3,22,23,24]. Prior research has described the single-stage posteriorly based dorsal tongue flap technique for moderate-sized (3–5 cm) intraoral defects near the RMT and lower alveolus, where bone exposure is a challenge [3]. Another study showcased the single-stage deep lingual artery axial propeller flap for various intraoral defects, ranging from moderate to large sizes (approximately 4.5 cm × 6.5 cm), encompassing lesions on the cheek, mouth floor, RMT, hard palate, and soft palate [10]. In our present study, we employed the posteriorly based pLTF technique, combined with the concept of a peninsular-shaped flap characterized by maintaining a minimal mucosal bridge at the pivot point and opening up mucosal barriers between the flap and the defect [2]. Our technique offers the advantage of addressing a broader spectrum of intraoral defects, including those involving the buccal mucosa, RMT, tongue root, and mouth floor, than previously mentioned studies. Furthermore, it is relatively simpler to execute since it eliminates the need for perforator dissection of the vascular pedicle [25,26,27]. It can be a time-consuming and intricate process commonly associated with creating whole island-shaped flaps.

Our technique hinges on two fundamental principles that facilitate reliable flap reconstruction with a single-stage operation. First, it affords additional vascular support and structural stability to the flap by maintaining a minimal mucosal bridge at the pivot point, reminiscent of the peninsular-shaped flap [2,18]. This innovation also reduces operation time by preventing the requirement for incisions and dissection of skin and tissue around this minimal mucosal bridge [2,18]. Second, the technique opens up the mucosal barrier between the flap and the defect, resulting in a single-stage flap surgery without requiring additional pedicle division procedures [2]. These two central tenets of the pLTF approach allowed for the safe completion of all flap surgeries within a short flap harvesting time (mean value, 28.75 ± 5.17 min). In addition, all flaps exhibited complete survival without any complications, and single-stage tongue flap reconstructions were accomplished without the need for supplementary operations. Furthermore, five patients underwent postoperative radiation therapy several months after the flap surgery, and no complications associated with radiation therapy, such as wound dehiscence, wound infection, flap necrosis, or fistula formation, were observed in the reconstructed areas with pLTFs. These findings suggest that the pLTF technique provides sufficient durability to withstand postoperative radiation therapy in intraoral reconstruction.

Despite these advantages and the utility of the pLTF, it has not been widely adopted as the primary choice for covering intraoral defects. This hesitance may be due to concerns about potential deficits in tongue functions, including chewing, swallowing, taste, and phonation, after partial tongue tissue is used as a flap [2]. Notably, several studies have already provided evidence to debunk such concerns, establishing that the application of the tongue flap does not result in significant functional deficits [3,5,28]. Consistent with these findings, our study demonstrated no statistically significant difference between the average pre- and postoperative FIGS scores.

However, it is essential to acknowledge the limitations of our study. The small sample size is the most notable limitation, rendering the study susceptible to bias and potentially undermining internal and external validity due to sensitivity to sample size. Moreover, this study is a retrospective case series lacking comparison groups, which may introduce selection bias if the choice between exposed and non-exposed subjects somehow correlates with the outcome. In addition, as an observational clinical study, certain confounding factors are unavoidable. Well-designed prospective studies with larger sample sizes and comparative groups are warranted to ensure the validity of our consistent results [29,30]. Nevertheless, our study carries significance as a consecutive case series showcasing single-stage pLTF reconstruction of various intraoral defects performed by a single surgeon, yielding favorable functional outcomes assessed using the FIGS. Furthermore, we have provided illustrative figures (Figure 1) to enhance reader comprehension of our single-stage pLTF technique.

## 5. Conclusions

Drawing from our study experience, we confidently propose the pLTF technique as a highly effective reconstructive modality for addressing a range of intraoral defects characterized by small to moderate sizes. This recommendation is grounded in several compelling factors:

First, the pLTF technique has technical simplicity and ease of execution. The pLTF technique offers a straightforward and manageable approach to intraoral defect coverage, making it accessible to a broader range of surgical teams. Compared with other techniques, its relative simplicity streamlines the surgical process.

Second, the pLTF has reliable flap survival. Our study results underscore the consistent and reliable survival of pLTFs in all cases, with no recorded instances of flap failure or postoperative complications. This dependable performance is a testament to the viability of the pLTF approach.

Third, the pLTF shows favorable functional outcomes. The preservation of good tongue function is paramount in intraoral defect reconstruction. Our findings, as evaluated by the FIGS, demonstrate that the pLTF technique successfully maintains or, in selected cases, even enhances tongue function.

Fourth, the pLTF can facilitate personalized intraoral reconstruction. The versatility of the pLTF technique allows for personalized intraoral reconstruction, accommodating a variety of defect sizes and locations. This adaptability ensures that each patient receives a tailored and effective solution.

Given these strengths, we confidently assert that the single-stage pLTF technique represents a valuable addition to the armamentarium of reconstructive options for addressing intraoral defects. Our study experience underscores the potential of pLTFs as a preferred choice in specific cases while acknowledging the need for further research and validation through larger-scale prospective studies. By embracing this technique, surgeons can offer their patients a customized and effective approach to intraoral defect coverage, ultimately contributing to improved patient outcomes and quality of life. The adoption of the pLTF technique has the potential to benefit patients by providing them with a customized and effective approach to intraoral defect coverage. Surgeons and clinicians should consider this technique as a viable option, particularly in cases where it aligns with the patient’s needs and the characteristics of the defect.

## Figures and Tables

**Figure 1 jpm-13-01637-f001:**
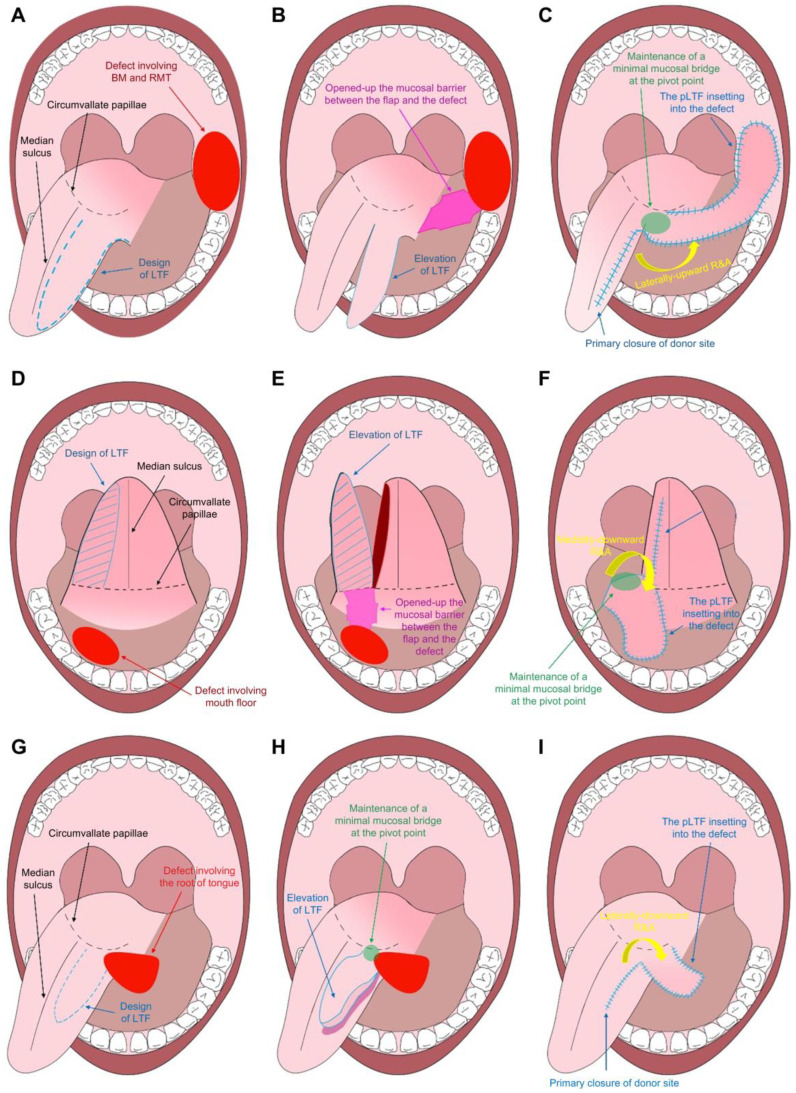
A schematic illustration of the single-stage peninsular-shaped lateral tongue flap (pLTF) reconstruction for intraoral defect coverage in various anatomical areas. (**A**–**C**) A single-stage pLTF reconstruction for covering a defect involving the buccal mucosa and retromolar trigone areas. (**D**–**F**) A single-stage pLTF reconstruction for covering a defect involving the mouth floor. (**G**–**I**) A single-stage pLTF reconstruction for covering a defect involving the root of the tongue. LTF, lateral tongue flap; BM, buccal mucosa; RMT, retromolar trigone; pLTF, peninsular-shaped lateral tongue flap; R&A, rotation and advancement.

**Figure 2 jpm-13-01637-f002:**
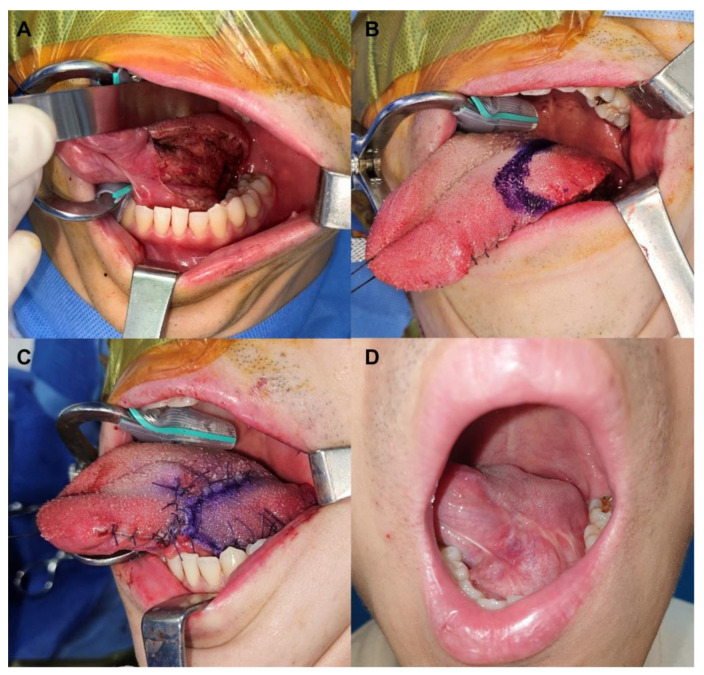
Clinical photographs of case 1. (**A**) An intraoral defect (3 cm × 4 cm) on the root of the tongue. (**B**) Design of a posteriorly based peninsular-shaped lateral tongue flap (pLTF) with a size of 3.5 cm × 5 cm at the ipsilateral side of the tongue. (**C**) Transfer of the flap to the defect through laterally downward rotation and advancement movement and successful coverage of the defect with the pLTF. (**D**) Postoperative photographs after a 7-month follow-up.

**Figure 3 jpm-13-01637-f003:**
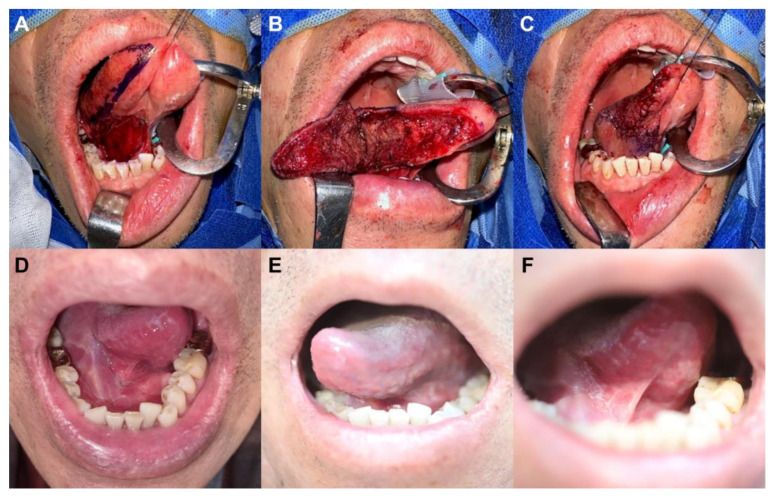
Clinical photographs of case 2. (**A**) An intraoral defect (3 cm × 4 cm) on the mouth floor. (**B**) Elevation of a posteriorly based peninsular-shaped lateral tongue flap (pLTF) with its size of 3 cm × 6 cm at the ipsilateral side of the tongue. (**C**) Transfer of the flap to the defect through medially downward rotation and advancement movement and successful coverage of the defect with the pLTF. (**D**–**F**) Postoperative photographs after a 7-month follow-up.

**Figure 4 jpm-13-01637-f004:**
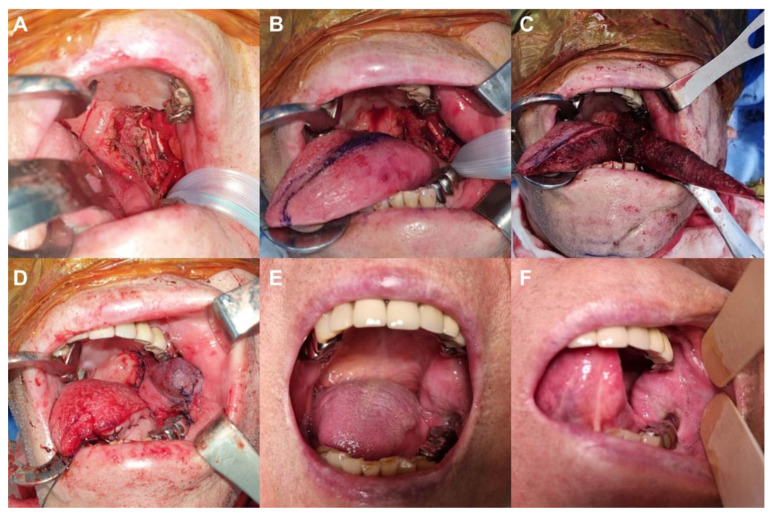
Clinical photographs of case 4. (**A**) An intraoral defect (3.5 cm × 4.5 cm) on the buccal mucosa and retromolar trigone area. (**B**) Design of a posteriorly based peninsular-shaped lateral tongue flap (pLTF) with a size of 4 cm × 6 cm at the ipsilateral side of the tongue. (**C**) Elevation of the flap. (**D**) Transfer of the flap to the defect through laterally upward rotation and advancement movement and successful coverage of the defect with the pLTF. (**E**,**F**) Postoperative photographs after a 15-month follow-up.

**Table 1 jpm-13-01637-t001:** Summary of patients’ data.

Case	Sex/Age (yrs)	Defect Cause	Defect Location	Defect Size (cm^2^)	Flap Size (cm^2^)	Flap Elevation Time (min)	Flap Survival	Postoperative Complication	Pathologic TNM Stage	Postoperative Radiation Therapy	Post-Radiation Therapy Complication	Follow-Up Period (Months)
1	M/34	SCC	ROT	3 × 4	3.5 × 5	25	Complete	None	pT1N0M0	None	None	7
2	M/62	AC	MF	3 × 4	3 × 6	25	Complete	None	pT1N0M0	60 Gy/30 fractions	None	7
3	F/41	SCC	ROT	3 × 3	3 × 4.5	20	Complete	None	pT1N0M0	55 Gy/30 fractions	None	8
4	M/86	SCC	BM and RMT	3.5 × 4.5	4 × 6	35	Complete	None	pT3N0M0	None	None	15
5	M/67	AC	MF	3 × 3.5	3.5 × 5	30	Complete	None	pT1N0M0	None	None	9
6	F/58	SCC	BM and RMT	4 × 6	4.5 × 7.5	35	Complete	None	pT1N0M0	54 Gy/30 fractions	None	12
7	M/75	ACC	BM and RMT	4 × 5	4.6 × 6.5	30	Complete	None	pT3N0M0	66 Gy/35 fractions	None	11
8	M/59	SCC	ROT	3 × 3	3 × 5	30	Complete	None	pT1N0M0	60 Gy/30 fractions	None	10

M, male; F, female; SCC, squamous cell carcinoma; AC, adenocarcinoma; ACC, adenocystic carcinoma; ROT, root of tongue; MF, mouth floor; BM, buccal mucosa; RMT, retromolar trigone; T, primary tumor; N, regional lymph node; M, distant metastasis; Gy, gray.

**Table 2 jpm-13-01637-t002:** Pre- and postoperative functional intraoral Glasgow scale (FIGS) scores.

Case	FIGS Score							
	Preop Chew	Preop Swallow	Preop Speak	Preop FIGS	Postop Chew	Postop Swallow	Postop Speak	Postop FIGS
1	5	5	5	15	5	5	5	15
2	5	5	5	15	5	5	4	14
3	5	5	5	15	5	5	5	15
4	4	5	5	14	5	5	4	14
5	5	5	5	15	5	5	4	14
6	5	5	5	15	5	5	4	14
7	4	5	5	14	4	4	4	12
8	5	5	5	15	5	5	4	14
Mean (*p*)				14.75 ± 0.46				14.00 ± 0.92 (0.059)

FIGS, functional intraoral Glasgow scale; Preop, preoperative; Postop, postoperative.

## Data Availability

The data presented in this study are available upon request from the corresponding author. The data are not publicly available due to privacy restrictions.

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
