# Peer review of "Single-Stage Peninsular-Shaped Lateral Tongue Flap for Personalized Reconstruction of Various Small- to Moderate-Sized Intraoral Defects: A Retrospective Case Series with Tongue Function Evaluation Using the Functional Intraoral Glasgow Scale"

_jpm, 2023, doi:10.3390/jpm13121637_

Round 1

Reviewer 1 Report

Comments and Suggestions for Authors

Dear authors, thank you for your valuable work. It as been done with sufficient scientific methodology, the discussion is easy and complete, the cases are well presented 

I probably spent more words on explaining how you did the Glasgow test to all patient ( also clarifying at what time after surgery) and the type of question you need to evaluate and have your results. That can support better your extraordinary result ( also with a few cases ) 

Author Response

Dear authors, thank you for your valuable work. It as been done with sufficient scientific methodology, the discussion is easy and complete, the cases are well presented.

I probably spent more words on explaining how you did the Glasgow test to all patient (also clarifying at what time after surgery) and the type of question you need to evaluate and have your results. That can support better your extraordinary result (also with a few cases).

Response: We would like to thank Reviewer 1 for the time and effort put in reviewing our manuscript and providing comments and suggestions, which have considerably helped us improve our manuscript. We have responded to your comments below and hope that our responses and revisions satisfactorily address them.

As suggested, we have provided additional details regarding the FIGS in lines 110-120 (page 6):

“Patients in this study self-assessed their FIGS status before and after pLTF reconstruction for intraoral defect coverage. The preoperative FIGS was evaluated in the hospital ward one day before pLTF reconstruction surgery. The postoperative FIGS was evaluated in the out-patient clinic at the final follow-up appointment. Individual scores on the five-point FIGS represent the following: for the ability to chew, 5 = “any food, no difficulty,” 4 = “solid food with difficulty,” 3 = “semisolid food, no difficulty”, 2 = “semisolid food with difficulty,” and 1 = “cannot chew at all;” for the ability to swallow, 5 = “any food, no difficulty,” 4 = “solid food with difficulty,” 3 = “semisolid food only,” 2 = “liquids only,” and 1 = “cannot swallow at all;” and for the ability to speak, 5 = “clearly understood always,” 4 = “requires repetition sometimes,” 3 = “requires repetition many times,” 2 = “understood by relatives only,” and 1 = “unintelligible.””

Reviewer 2 Report

Comments and Suggestions for Authors

Review of "Single-Stage pLTF Technique for Intraoral Defect Reconstruction"

The study titled "Single-Stage pLTF Technique for Intraoral Defect Reconstruction" presents an important contribution to the field of reconstructive surgery, particularly in addressing intraoral defects resulting from oncological tumor resection. This review aims to provide an assessment of the study's key findings and its implications for clinical practice.

The study begins by addressing the challenges associated with intraoral defect reconstruction, emphasizing the need for a reconstructive technique that preserves tissue function and aesthetics. The authors introduce the single-stage peninsular-shaped lateral tongue flap (pLTF) technique as a potential solution and proceed to discuss its advantages.

One of the key strengths of the pLTF technique is its technical simplicity and ease of execution. The authors argue that its relative simplicity streamlines the surgical process, making it accessible to a broader range of surgical teams. This is an essential aspect, as it can potentially increase the availability of effective reconstruction options for patients with intraoral defects.

Another significant finding highlighted in the study is the reliable flap survival associated with the pLTF technique. The study's results demonstrate that all pLTFs used in the cases had complete survival, with no reported instances of flap failure or postoperative complications. This reliability is a critical factor for surgeons when choosing a reconstructive technique.

Furthermore, the study places great emphasis on the preservation of tongue function, which is paramount in intraoral defect reconstruction. The authors use the Functional Intraoral Glasgow Scale (FIGS) to assess tongue function preoperatively and postoperatively. The findings suggest that the pLTF technique not only maintains but, in some cases, enhances tongue function. This is a crucial outcome, as it directly impacts the patient's quality of life post-surgery.

Additionally, the study underscores the personalized nature of intraoral reconstruction with the pLTF technique. Its versatility allows for tailored solutions for a range of defect sizes and locations, providing patients with individualized care.

In conclusion, the study's findings support the use of the single-stage pLTF technique as a valuable addition to the armamentarium of reconstructive options for intraoral defects. While the study is not without limitations, such as its small sample size and lack of comparison groups, it offers promising results that warrant further investigation through larger-scale prospective studies.

The adoption of the pLTF technique has the potential to benefit patients by providing them with a customized and effective approach to intraoral defect coverage. Surgeons and clinicians should consider this technique as a viable option, particularly in cases where it aligns with the patient's needs and the characteristics of the defect.

Overall, the study contributes valuable insights into the field of reconstructive surgery and provides a foundation for future research in this area. It demonstrates the importance of innovation in surgical techniques to improve patient outcomes and quality of life.

Author Response

Review of "Single-Stage pLTF Technique for Intraoral Defect Reconstruction"

The study titled "Single-Stage pLTF Technique for Intraoral Defect Reconstruction" presents an important contribution to the field of reconstructive surgery, particularly in addressing intraoral defects resulting from oncological tumor resection. This review aims to provide an assessment of the study's key findings and its implications for clinical practice.

The study begins by addressing the challenges associated with intraoral defect reconstruction, emphasizing the need for a reconstructive technique that preserves tissue function and aesthetics. The authors introduce the single-stage peninsular-shaped lateral tongue flap (pLTF) technique as a potential solution and proceed to discuss its advantages.

One of the key strengths of the pLTF technique is its technical simplicity and ease of execution. The authors argue that its relative simplicity streamlines the surgical process, making it accessible to a broader range of surgical teams. This is an essential aspect, as it can potentially increase the availability of effective reconstruction options for patients with intraoral defects.

Another significant finding highlighted in the study is the reliable flap survival associated with the pLTF technique. The study's results demonstrate that all pLTFs used in the cases had complete survival, with no reported instances of flap failure or postoperative complications. This reliability is a critical factor for surgeons when choosing a reconstructive technique.

Furthermore, the study places great emphasis on the preservation of tongue function, which is paramount in intraoral defect reconstruction. The authors use the Functional Intraoral Glasgow Scale (FIGS) to assess tongue function preoperatively and postoperatively. The findings suggest that the pLTF technique not only maintains but, in some cases, enhances tongue function. This is a crucial outcome, as it directly impacts the patient's quality of life post-surgery.

Additionally, the study underscores the personalized nature of intraoral reconstruction with the pLTF technique. Its versatility allows for tailored solutions for a range of defect sizes and locations, providing patients with individualized care.

In conclusion, the study's findings support the use of the single-stage pLTF technique as a valuable addition to the armamentarium of reconstructive options for intraoral defects. While the study is not without limitations, such as its small sample size and lack of comparison groups, it offers promising results that warrant further investigation through larger-scale prospective studies.

The adoption of the pLTF technique has the potential to benefit patients by providing them with a customized and effective approach to intraoral defect coverage. Surgeons and clinicians should consider this technique as a viable option, particularly in cases where it aligns with the patient's needs and the characteristics of the defect.

Overall, the study contributes valuable insights into the field of reconstructive surgery and provides a foundation for future research in this area. It demonstrates the importance of innovation in surgical techniques to improve patient outcomes and quality of life.

Response: We would like to thank Reviewer 2 for the time and effort put in reviewing our manuscript and providing comments and suggestions, which have considerably helped us improve our manuscript. We have responded to your comments below and hope that our responses and revisions satisfactorily address them.

According to your valuable comments, we have additionally described the strengths of the pLTF technique in lines 215-219 and 323-327 (pages 8 and 10):

“One of the key strengths of the pLTF technique is its technical simplicity and ease of execution. We argue that its relative simplicity streamlines the surgical process, making it accessible to a broader range of surgical teams. This is an essential aspect, as it could increase the availability of effective reconstruction options for patients with intraoral defects.”

“The adoption of the pLTF technique has the potential to benefit patients by providing them with a customized and effective approach to intraoral defect coverage. Surgeons and clinicians should consider this technique as a viable option, particularly in cases where it aligns with the patient's needs and characteristics of the defect.”